# Generative Probabilistic Novelty Detection with Adversarial Autoencoders

**Stanislav Pidhorskyi**    **Ranya Almohsen**    **Donald A. Adjeroh**    **Gianfranco Doretto**

Lane Department of Computer Science and Electrical Engineering
West Virginia University, Morgantown, WV 26506
`{stpidhorskyi, ralmohse, daadjeroh, gidoretto}@mix.wvu.edu`

## Abstract

Novelty detection is the problem of identifying whether a new data point is considered to be an inlier or an outlier. We assume that training data is available to describe only the inlier distribution. Recent approaches primarily leverage deep encoder-decoder network architectures to compute a reconstruction error that is used to either compute a novelty score or to train a one-class classifier. While we too leverage a novel network of that kind, we take a probabilistic approach and effectively compute how likely it is that a sample was generated by the inlier distribution. We achieve this with two main contributions. First, we make the computation of the novelty probability feasible because we linearize the parameterized manifold capturing the underlying structure of the inlier distribution, and show how the probability factorizes and can be computed with respect to local coordinates of the manifold tangent space. Second, we improve the training of the autoencoder network. An extensive set of results show that the approach achieves state-of-the-art performance on several benchmark datasets.

## 1 Introduction

Novelty detection is the problem of identifying whether a new data point is considered to be an *inlier* or an *outlier*. From a statistical point of view this process usually occurs while prior knowledge of the distribution of inliers is the only information available. This is also the most difficult and relevant scenario because outliers are often very rare, or even dangerous to experience (e.g., in industry process fault detection [1]), and there is a need to rely only on inlier training data. Novelty detection has received significant attention in application areas such as medical diagnoses [2], drug discovery [3], and among others, several computer vision applications, such as anomaly detection in images [4, 5], videos [6], and outlier detection [7, 8]. We refer to [9] for a general review on novelty detection. The most recent approaches are based on learning deep network architectures [10, 11], and they tend to either learn a one-class classifier [12, 11], or to somehow leverage as novelty score, the reconstruction error of the encoder-decoder architecture they are based on [13, 7].

In this work, we introduce a new encoder-decoder architecture as well, which is based on adversarial autoencoders [14]. However, we do not train a one-class classifier, instead, we learn the probability distribution of the inliers. Therefore, the novelty test simply becomes the evaluation of the probability of a test sample, and rare samples (outliers) fall below a given threshold. We show that this approach allows us to effectively use the decoder network to learn the parameterized manifold shaping the inlier distribution, in conjunction with the probability distribution of the (parameterizing) latent space. The approach is made computationally feasible because for a given test sample we linearize the manifold, and show that with respect to the local manifold coordinates the data model distribution factorizes into a component dependent on the manifold (decoder network plus latent distribution), and another one dependent on the noise, which can also be learned offline.

We named the approach *generative probabilistic novelty detection (GPND)* because we compute the probability distribution of the full model, which includes the signal plus noise portion, and because it relies on being able to also generate data samples. We are mostly concerned with novelty detection using images, and with controlling the distribution of the latent space to ensure good generative reproduction of the inlier distribution. This is essential not so much to ensure good image generation, but for the correct computation of the novelty score. This aspect has been overlooked by the deep learning literature so far, since the focus has been only on leveraging the reconstruction error. We do leverage that as well, but we show in our framework that the reconstruction error affects only the noise portion of the model. In order to control the latent distribution and image generation we learn an adversarial autoencoder network with two discriminators that address these two issues.

Section 2 reviews the related work. Section 3 introduces the GPND framework, and Section 4 describes the training and architecture of the adversarial autoencoder network. Section 6 shows a rich set of experiments showing that GPND is very effective and produces state-of-the-art results on several benchmarks.

## 2 Related Work

Novelty detection is the task of recognizing abnormality in data. The literature in this area is sizable. Novelty detection methods can be statistical and probabilistic based [15, 16], distance based [17], and also based on self-representation [8]. Recently, deep learning approaches [7, 11] have also been used, greatly improving the performance of novelty detection.

Statistical methods [18, 19, 15, 16] usually focus on modeling the distribution of inliers by learning the parameters defining the probability, and outliers are identified as those having low probability under the learned model. Distance based outlier detection methods [20, 17, 21] identify outliers by their distance to neighboring examples. They assume that inliers are close to each other while the abnormal samples are far from their nearest neighbors. A known work in this category is LOF [22], which is based on $k$-nearest neighbors and density based estimation. More recently, [23] introduced the Kernel Null Foley-Sammon Transform (KNFST) for multi-class novelty detection, where training samples of each known category are projected onto a single point in the null space and then distances between the projection of a test sample and the class representatives are used to obtain a novelty measure. [24] improves on previous approaches by proposing an incremental procedure called Incremental Kernel Null Space Based Discriminant Analysis (IKNDA).

Since outliers do not have sparse representations, self-representation approaches have been proposed for outlier detection in a union of subspaces [4, 25]. Similarly, deep learning based approaches have used neural networks and leveraged the reconstruction error of encoder-decoder architectures. [26, 27] used deep learning based autoencoders to learn the model of normal behaviors and employed a reconstruction loss to detect outliers. [28] used a GAN [29] based method by generating new samples similar to the training data, and demonstrated its ability to describe the training data. Then it transformed the implicit data description of normal data to a novelty score. [10] trained GANs using optical flow images to learn a representation of scenes in videos. [7] minimized the reconstruction error of an autoencoder to remove outliers from noisy data, and by utilizing the gradient magnitude of the auto-encoder they make the reconstruction error more discriminative for positive samples. In [11] they proposed a framework for one-class classification and novelty detection. It consists of two main modules learned in an adversarial fashion. The first is a decoder-encoder convolutional neural network trained to reconstruct inliers accurately, while the second is a one-class classifier made with another network that produces the novelty score.

The proposed approach relates to the statistical methods because it aims at computing the probability distribution of test samples as novelty score, but it does so by learning the manifold structure of the distribution with an encoder-decoder network. Moreover, the method is different from those that learn a one-class classifier, or rely on the reconstruction error to compute the novelty score, because in our framework we represent only one component of the score computation, allowing to achieve an improved performance.

State-of-the art works on density estimation for image compression include Pixel Recurrent Neural Networks [30] and derivatives [31, 32]. These pixel-based methods allow to sequentially predict pixels in an image along the two spatial dimensions. Because they model the joint distribution of the raw pixels along with their sequential correlation, it is possible to use them for image compression.

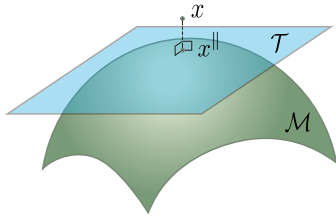

Figure 1: **Manifold schematic representation.** This figure shows connection between the parametrized manifold $\mathcal{M}$, its tangent space $\mathcal{T}$, data point $x$ and its projection $x^{\parallel}$.

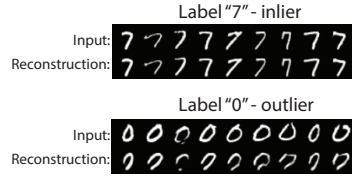

Figure 2: **Reconstruction of inliers and outliers.** This figure showns reconstructions for the autoencoder network that was trained on inlier of label "7" of MNIST [37] dataset. First line is input of inliers of label "7", the second line shows corresponding reconstructions. The third line corresponds to input of outlier of label "0" and the forth line, corresponding reconstructions.

Although they could also model the probability distribution of known samples, they work at a local scale in a patch-based fashion, which makes non-local pixels loosely correlated. Our approach instead, does not allow modeling the probability density of individual pixels but works with the whole image. It is not suitable for image compression, and while its generative nature allows in principle to produce novel images, in this work we focus only on novelty detection by evaluating the inlier probability distribution on test samples.

A recent line of work has focussed on detecting out-of-distribution samples by analyzing the output entropy of a prediction made by a pre-trained deep neural network [33, 34, 35, 36]. This is done by either simply thresholding the maximum softmax score [34], or by first applying perturbations to the input, scaled proportionally to the gradients w.r.t. to the input and then combining the softmax score with temperature scaling, as it is done in Out-of-distribution Image Detection in Neural Networks (ODIN) [36]. While these approaches require labels for the in-distribution data to train the classifier network, our method does not use label information. Therefore, it can be applied for the case when in-distribution data is represented by one class or label information is not available.

## 3 Generative Probabilistic Novelty Detection

We assume that training data points $x_1, \ldots, x_N$, where $x_i \in \mathbb{R}^m$, are sampled, possibly with noise $\xi_i$, from the model

$$x_i = f(z_i) + \xi_i \qquad i = 1, \cdots, N \ , \tag{1}$$

where $z_i \in \Omega \subset \mathbb{R}^n$. The mapping $f : \Omega \to \mathbb{R}^m$ defines $\mathcal{M} \equiv f(\Omega)$, which is a parameterized manifold of dimension $n$, with $n < m$. We also assume that the Jacobi matrix of $f$ is full rank at every point of the manifold. In addition, we assume that there is another mapping $g : \mathbb{R}^m \to \mathbb{R}^n$, such that for every $x \in \mathcal{M}$, it follows that $f(g(x)) = x$, which means that $g$ acts as the inverse of $f$ on such points.

Given a new data point $\bar{x} \in \mathbb{R}^m$, we design a novelty test to assert whether $\bar{x}$ was sampled from model (1). We begin by observing that $\bar{x}$ can be non-linearly projected onto $\bar{x}^{\parallel} \in \mathcal{M}$ via $\bar{x}^{\parallel} = f(\bar{z})$, where $\bar{z} = g(\bar{x})$. Assuming $f$ to be smooth enough, we perform a linearization based on its first-order Taylor expansion

$$f(z) = f(\bar{z}) + J_f(\bar{z})(z - \bar{z}) + O(\|z - \bar{z}\|^2) \ , \tag{2}$$

where $J_f(\bar{z})$ is the Jacobi matrix computed at $\bar{z}$, and $\| \cdot \|$ is the L$_2$ norm. We note that $\mathcal{T} = \text{span}(J_f(\bar{z}))$ represents the tangent space of $f$ at $\bar{x}^{\parallel}$ that is spanned by the $n$ independent column vectors of $J_f(\bar{z})$, see Figure 1. Also, we have $\mathcal{T} = \text{span}(U^{\parallel})$, where $J_f(\bar{z}) = U^{\parallel} S V^{\top}$ is the singular value decomposition (SVD) of the Jacobi matrix. The matrix $U^{\parallel}$ has rank $n$, and if we define $U^{\perp}$ such that $U = [U^{\parallel} U^{\perp}]$ is a unitary matrix, we can represent the data point $\bar{x}$ with respect to the local coordinates that define the tangent space $\mathcal{T}$, and its orthogonal complement $\mathcal{T}^{\perp}$. This is done

by computing

$$\bar{w} = U^\top \bar{x} = \left[ \begin{array}{c} {U^\parallel}^\top \bar{x} \\ {U^\perp}^\top \bar{x} \end{array} \right] = \left[ \begin{array}{c} \bar{w}^\parallel \\ \bar{w}^\perp \end{array} \right] , \tag{3}$$

where the rotated coordinates $\bar{w}$ are decomposed into $\bar{w}^\parallel$, which are parallel to $\mathcal{T}$, and $\bar{w}^\perp$ which are orthogonal to $\mathcal{T}$.

We now indicate with $p_X(x)$ the probability density function describing the random variable $X$, from which training data points have been drawn. Also, $p_W(w)$ is the probability density function of the random variable $W$ representing $X$ after the change of coordinates. The two distributions are identical. However, we make the assumption that the coordinates $W^\parallel$, which are parallel to $\mathcal{T}$, and the coordinates $W^\perp$, which are orthogonal to $\mathcal{T}$, are statistically independent. This means that the following holds

$$p_X(x) = p_W(w) = p_W(w^\parallel, w^\perp) = p_{W^\parallel}(w^\parallel) p_{W^\perp}(w^\perp) . \tag{4}$$

This is motivated by the fact that in (1) the noise $\xi$ is assumed to predominantly deviate the point $x$ away from the manifold $\mathcal{M}$ in a direction orthogonal to $\mathcal{T}$. This means that $W^\perp$ is primarily responsible for the noise effects, and since noise and drawing from the manifold are statistically independent, so are $W^\parallel$ and $W^\perp$.

From (4), given a new data point $\bar{x}$, we propose to perform novelty detection by executing the following test

$$p_X(\bar{x}) = p_{W^\parallel}(\bar{w}^\parallel) p_{W^\perp}(\bar{w}^\perp) = \left\{ \begin{array}{lll} \geq \gamma & \implies & \text{Inlier} \\ < \gamma & \implies & \text{Outlier} \end{array} \right. \tag{5}$$

where $\gamma$ is a suitable threshold.

## 3.1   Computing the distribution of data samples

The novelty detector (5) requires the computation of $p_{W^\parallel}(w^\parallel)$ and $p_{W^\perp}(w^\perp)$. Given a test data point $\bar{x} \in \mathbb{R}^m$ its non-linear projection onto $\mathcal{M}$ is $\bar{x}^\parallel = f(g(\bar{x}))$. Therefore, $\bar{w}^\parallel$ can be written as $\bar{w}^\parallel = {U^\parallel}^\top \bar{x} = {U^\parallel}^\top (\bar{x} - \bar{x}^\parallel) + {U^\parallel}^\top \bar{x}^\parallel = {U^\parallel}^\top \bar{x}^\parallel$, where we have made the approximation that ${U^\parallel}^\top (\bar{x} - \bar{x}^\parallel) \approx 0$. Since $\bar{x}^\parallel \in \mathcal{M}$, then in its neighborhood it can be parameterized as in (2), which means that $w^\parallel(z) = {U^\parallel}^\top f(\bar{z}) + SV^\top(z - \bar{z}) + O(\|z - \bar{z}\|^2)$. Therefore, if $Z$ represents the random variable from which samples are drawn from the parameterized manifold, and $p_Z(z)$ is its probability density function, then it follows that

$$p_{W^\parallel}(w^\parallel) = |\det S^{-1}| p_Z(z) , \tag{6}$$

since $V$ is a unitary matrix. We note that $p_Z(z)$ is a quantity that is independent from the linearization (2), and therefore it can be learned offline, as explained in Section 5.

In order to compute $p_{W^\perp}(w^\perp)$, we approximate it with its average over the hypersphere $\mathcal{S}^{m-n-1}$ of radius $\|w^\perp\|$, giving rise to

$$p_{W^\perp}(w^\perp) \approx \frac{\Gamma\left(\frac{m-n}{2}\right)}{2\pi^{\frac{m-n}{2}} \|w^\perp\|^{m-n}} p_{\|W^\perp\|}(\|w^\perp\|) , \tag{7}$$

where $\Gamma(\cdot)$ represents the gamma function. This is motivated by the fact that noise of a given intensity will be equally present in every direction. Moreover, its computation depends on $p_{\|W^\perp\|}(\|w^\perp\|)$, which is the distribution of the norms of $w^\perp$, and which can easily be learned offline by histogramming the norms of $\bar{w}^\perp = {U^\perp}^\top \bar{x}$.

## 4   Manifold learning with adversarial autoencoders

In this section we describe the network architecture and the training procedure for learning the mapping $f$ that define the parameterized manifold $\mathcal{M}$, and also the mapping $g$. The mappings $g$ and $f$ represent and are modeled by an *encoder* network, and a *decoder* network, respectively.

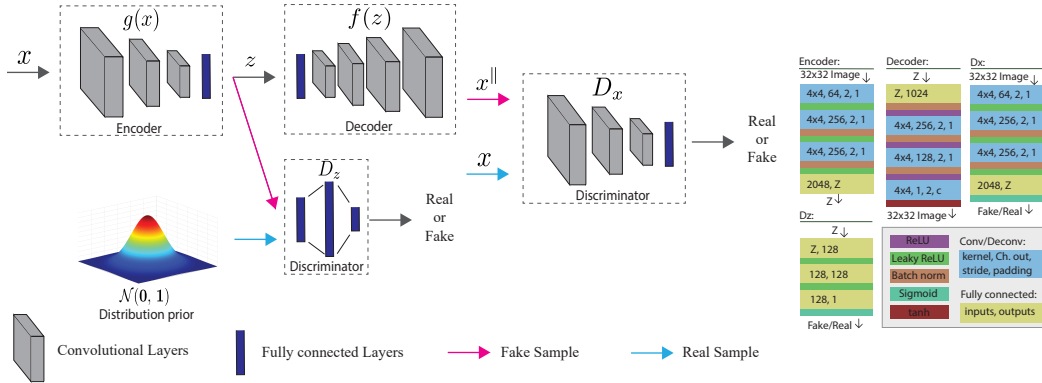

Figure 3: **Architecture overview.** Architecture of the network for manifold learning. It is based on training an Adversarial Autoenconder (AAE) [14]. Similarly to [43, 11] it has an additional adversarial component to improve generative capabilities of decoded images and a better manifold learning. The architecture layers of the AAE and of the discriminator $D_x$ are specified on the right.

Similarly to previous work on novelty detection [38, 39, 40, 7, 11, 13], such networks are based on autoencoders [41, 42].

The autoencoder network and training should be such that they reproduce the manifold $\mathcal{M}$ as closely as possible. For instance, if $\mathcal{M}$ represents the distribution of images depicting a certain object category, we would want the estimated encoder and decoder to be able to generate images as if they were drawn from the real distribution. Differently from previous work, we require the latent space, represented by $z$, to be close to a known distribution, preferably a normal distribution, and we would also want each of the components of $z$ to be maximally informative, which is why we require them to be independent random variables. Doing so facilitates learning a distribution $p_Z(z)$ from training data mapped onto the latent space $\Omega$. This means that the autoenoder has generative properties, because by sampling from $p_Z(z)$ we would generate data points $x \in \mathcal{M}$. Note that differently from GANs [29] we also require an encoder function $g$.

Variational Auto-Encoders (VAEs) [44] are known to work well in presence of continuous latent variables and they can generate data from a randomly sampled latent space. VAEs utilize stochastic variational inference and minimize the Kullback-Leibler (KL) divergence penalty to impose a prior distribution on the latent space that encourages the encoder to learn the modes of the prior distribution. Adversarial Autoencoders (AAEs) [14], in contrast to VAEs, use an adversarial training paradigm to match the posterior distribution of the latent space with the given distribution. One of the advantages of AAEs over VAEs is that the adversarial training procedure encourages the encoder to match the whole distribution of the prior.

Unfortunately, since we are concerned with working with images, both AAEs and VAEs tend to produce examples that are often far from the real data manifold. This is because the decoder part of the network is updated only from a reconstruction loss that is typically a pixel-wise cross-entropy between input and output image. Such loss often causes the generated images to be blurry, which has a negative effect on the proposed approach. Similarly to AAEs, PixelGAN autoencoders [45] introduce the adversarial component to impose a prior distribution on the latent code, but the architecture is significantly different, since it is conditioned on the latent code.

Similarly to [43, 11] we add an adversarial training criterion to match the output of the decoder with the distribution of real data. This allows to reduce blurriness and add more local details to the generated images. Moreover, we also combine the adversarial training criterion with AAEs, which results in having two adversarial losses: one to impose a prior on the latent space distribution, and the second one to impose a prior on the output distribution.

Our full objective consists of three terms. First, we use an adversarial loss for matching the distribution of the latent space with the prior distribution, which is a normal with 0 mean, and standard deviation 1, $\mathcal{N}(0, 1)$. Second, we use an adversarial loss for matching the distribution of the decoded images

from $z$ and the known, training data distribution. Third, we use an autoencoder loss between the decoded images and the encoded input image. Figure 3 shows the architecture configuration.

## 4.1 Adversarial losses

For the discriminator $D_z$, we use the following adversarial loss:

$$\mathcal{L}_{adv-d_z}(x, g, D_z) = E[\log(D_z(\mathcal{N}(0, 1)))] + E[\log(1 - D_z(g(x)))] , \qquad (8)$$

where the encoder $g$ tries to encode $x$ to a $z$ with distribution close to $\mathcal{N}(0, 1)$. $D_z$ aims to distinguish between the encoding produced by $g$ and the prior normal distribution. Hence, $g$ tries to minimize this objective against an adversary $D_z$ that tries to maximize it.

Similarly, we add the adversarial loss for the discriminator $D_x$:

$$\mathcal{L}_{adv-d_x}(x, D_x, f) = E[\log(D_x(x))] + E[\log(1 - D_x(f(\mathcal{N}(0, 1))))] , \qquad (9)$$

where the decoder $f$ tries to generate $x$ from a normal distribution $\mathcal{N}(0, 1)$, in a way that $x$ is as if it was sampled from the real distribution. $D_x$ aims to distinguish between the decoding generated by $f$ and the real data points $x$. Hence, $f$ tries to minimize this objective against an adversary $D_x$ that tries to maximize it.

## 4.2 Autoencoder loss

We also optimize jointly the encoder $g$ and the decoder $f$ so that we minimize the reconstruction error for the input $x$ that belongs to the known data distribution.

$$\mathcal{L}_{error}(x, g, f) = -E_z[\log(p(f(g(x))|x))] , \qquad (10)$$

where $\mathcal{L}_{error}$ is minus the expected log-likelihood, i.e., the reconstruction error. This loss does not have an adversarial component but it is essential to train an autoencoder. By minimizing this loss we encourage $g$ and $f$ to better approximate the real manifold.

## 4.3 Full objective

The combination of all the previous losses gives

$$\mathcal{L}(x, g, D_z, D_x, f) = \mathcal{L}_{adv-d_z}(x, g, D_z) + \mathcal{L}_{adv-d_x}(x, D_x, f) + \lambda \mathcal{L}_{error}(x, g, f) , \qquad (11)$$

Where $\lambda$ is a parameter that strikes a balance between the reconstruction and the other losses. The autoencoder network is obtained by minimizing (11), giving:

$$\hat{g}, \hat{f} = \arg\min_{g,f} \max_{D_x, D_z} \mathcal{L}(x, g, D_z, D_x, f) . \qquad (12)$$

The model is trained using stochastic gradient descent by doing alternative updates of each component as follows

- Maximize $\mathcal{L}_{adv-d_x}$ by updating weights of $D_x$;
- Minimize $\mathcal{L}_{adv-d_x}$ by updating weights of $f$;
- Maximize $\mathcal{L}_{adv-d_z}$ by updating weights of $D_z$;
- Minimize $\mathcal{L}_{error}$ and $\mathcal{L}_{adv-d_z}$ by updating weights of $g$ and $f$.

# 5 Implementation Details and Complexity

After learning the encoder and decoder networks, by mapping the training set onto the latent space through $g$, we fit to the data a generalized Gaussian distribution and estimate $p_Z(z)$. In addition, by histogramming the quantities $\|U^{\perp^\top}(x - x^\|)\|$ we estimate $p_{\|W^\perp\|}(\|w^\perp\|)$. The entire training procedure takes about one hour with a high-end PC with one NVIDIA TITAN X.

When a sample is tested, the procedure entails mainly computing a derivative, i.e. the Jacoby matrix $J_f$, with a subsequent SVD. $J_f$ is computed numerically, around the test sample representation $\bar{z}$ and takes approximately 20.4ms for an individual sample and 0.55ms if computed as part of a batch of size 512, while the SVD takes approximately 4.0ms.

Table 1: $F_1$ scores on MNIST [37]. Inliers are taken to be images of one category, and outliers are randomly chosen from other categories.

| % of outliers | $\mathcal{D}(\mathcal{R}(X))$ [11] | $\mathcal{D}(X)$ [11] | LOF [22] | DRAE [7] | GPND (Ours) |
|---|---|---|---|---|---|
| 10 | 0.97 | 0.93 | 0.92 | 0.95 | **0.983** |
| 20 | 0.92 | 0.90 | 0.83 | 0.91 | **0.971** |
| 30 | 0.92 | 0.87 | 0.72 | 0.88 | **0.961** |
| 40 | 0.91 | 0.84 | 0.65 | 0.82 | **0.950** |
| 50 | 0.88 | 0.82 | 0.55 | 0.73 | **0.939** |

# 6 Experiments

We evaluate our novelty detection approach, which we call *Generative Probabilistic Novelty Detection (GPND)*, against several state-of-the-art approaches and with several performance measures. We use the $F_1$ measure, the area under the ROC curve (AUROC), the FPR at 95% TPR (i.e., the probability of an outlier to be misclassified as inlier), the Detection Error (i.e., the misclassification probability when TPR is 95%), and the area under the precision-recall curve (AUPR) when inliers (AUPR-In) or outliers (AUPR-Out) are specified as positives. All reported results are from our publicly available implementation[1], based on the deep machine learning framework PyTorch [46]. An overview of the architecture is provided in Figure 3.

## 6.1 Datasets

We evaluate GPND on the following datasets.
**MNIST** [37] contains $70,000$ handwritten digits from 0 to 9. Each of ten categories is used as inlier class and the rest of the categories are used as outliers.
**The Coil-100** dataset [47] contains $7,200$ images of 100 different objects. Each object has 72 images taken at pose intervals of 5 degrees. We downscale the images to size $32 \times 32$. We take randomly $n$ categories, where $n \in 1, 4, 7$ and randomly sample the rest of the categories for outliers. We repeat this procedure 30 times.
**Fashion-MNIST** [48] is a new dataset comprising of $28 \times 28$ grayscale images of $70,000$ fashion products from 10 categories, with $7,000$ images per category. The training set has $60,000$ images and the test set has $10,000$ images. Fashion-MNIST shares the same image size, data format and the structure of training and testing splits with the original MNIST.
**Others.** We compare GPND with ODIN [36] using their protocol. For inliers are used samples from CIFAR-10(CIFAR-100) [49], which is a publicly available dataset of small images of size $32 \times 32$, which have each been labeled to one of 10 (100) classes. Each class is represented by $6,000$ (600) images for a total of $60,000$ samples. For outliers are used samples from TinyImageNet [50], LSUN [51], and iSUN [52]. For more details please refer to [36]. We reuse the prepared datasets of outliers provided by the ODIN GitHub project page.

## 6.2 Results

**MNIST dataset.** We follow the protocol described in [11, 7] with some differences discussed below. Results are averages from a 5-fold cross-validation. Each fold takes 20% of each class. 60% of each class is used for training, 20% for validation, and 20% for testing. Once $p_X(\bar{x})$ is computed for each validation sample, we search for the $\gamma$ that gives the highest $F_1$ measure. For each class of digit, we train the proposed model and simulate outliers as randomly sampled images from other categories with proportion from 10% to 50%. Results for $\mathcal{D}(\mathcal{R}(X))$ and $\mathcal{D}(X)$ reported in [11] correspond to the protocol for which data is not split into separate training, validation and testing sets, meaning that the same inliers used for training were also used for testing. We diverge from this protocol and do not reuse the same inliers for training and testing. We follow the $60\%/20\%/20\%$ splits for training, validation and testing. This makes our testing harder, but more realistic, while we still compare our numbers against those obtained by others with easier settings. Results on the MNIST dataset are shown in Table 1 and Figure 4, where we compare with [11, 22, 7].

Table 2: Results on Coil-100. Inliers are taken to be images of one, four, or seven randomly chosen categories, and outliers are randomly chosen from other categories (at most one from each category).

| | OutRank [55, 56] | CoP [57] | REAPER [58] | OutlierPursuit [59] | LRR [60] | DPCP [61] | $\ell_1$ thresholding [25] | R-graph [8] | **Ours** |
|---|---|---|---|---|---|---|---|---|---|
| | | | | Inliers: **one** category of images , Outliers: 50% | | | | | |
| AUC | 0.836 | 0.843 | 0.900 | 0.908 | 0.847 | 0.900 | 0.991 | **0.997** | 0.968 |
| F1 | 0.862 | 0.866 | 0.892 | 0.902 | 0.872 | 0.882 | 0.978 | **0.990** | 0.979 |
| | | | | Inliers: **four** category of images , Outliers: 25% | | | | | |
| AUC | 0.613 | 0.628 | 0.877 | 0.837 | 0.687 | 0.859 | 0.992 | **0.996** | 0.945 |
| F1 | 0.491 | 0.500 | 0.703 | 0.686 | 0.541 | 0.684 | 0.941 | **0.970** | 0.960 |
| | | | | Inliers: **seven** category of images , Outliers: 15% | | | | | |
| AUC | 0.570 | 0.580 | 0.824 | 0.822 | 0.628 | 0.804 | 0.991 | **0.996** | 0.919 |
| F1 | 0.342 | 0.346 | 0.541 | 0.528 | 0.366 | 0.511 | 0.897 | **0.955** | 0.941 |

Table 3: Results on Fashion-MNIST [48]. Inliers are taken to be images of one category, and outliers are randomly chosen from other categories.

| % of outliers | 10 | 20 | 30 | 40 | 50 |
|---|---|---|---|---|---|
| F1 | 0.968 | 0.945 | 0.917 | 0.891 | 0.864 |
| AUC | 0.928 | 0.932 | 0.933 | 0.933 | 0.933 |

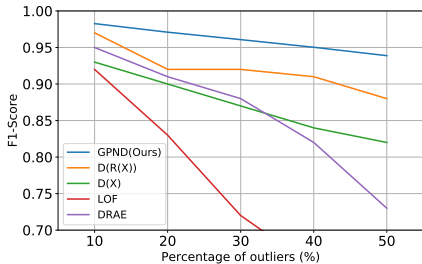

Figure 4: Results on MNIST [37] dataset.

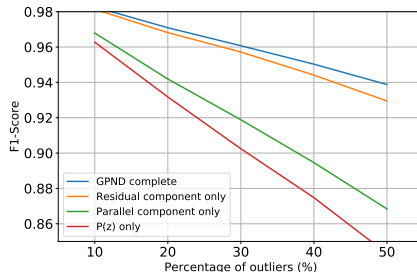

Figure 5: **Ablation study.** Comparison on MNIST of the model components of GPND.

**Coil-100 dataset.** We follow the protocol described in [8] with some differences discussed below. Results are averages from 5-fold cross-validation. Each fold takes 20% of each class. Because the count of samples per category is very small, we use 80% of each class for training, and 20% for testing. We find the optimal threshold $\gamma$ on the training set. Results reported in [8] correspond to not splitting data into separate training, validation and testing sets, because it is not essential, since they leverage a VGG [53] network pretrained on ImageNet [54]. We diverge from that protocol and do not reuse inliers and follow 80%/20% splits for training and testing.

Results on Coil-100 are shown in Table 2. We do not outperform R-graph [8], however as mentioned before, R-graph uses a pretrained VGG network, while we train an autoencoder from scratch on a very limited number of samples, which is on average only 70 per category.

**Fashion-MNIST dataset.** We repeat the same experiment with the same protocol that we have used for MNIST, but on Fashion-MNIST. Results are provided in Table 3.

**CIFAR-10 (CIFAR-100) dataset.** We follow the protocol described in [36], where for inliers and outliers are used different datasets. ODIN relies on a pretrained classifier and thus requires label information provided with the training samples, while our approach does not use label information. The results are reported in Table 4. Despite the fact that ODIN relies upon powerful classifier networks such as Dense-BC and WRN with more than 100 layers, the much smaller network of GPND competes well with ODIN. Note that for CIFAR-100, GPND significantly outperforms both ODIN architectures. We think this might be due to the fact that ODIN relies on the perturbation of the network classifier output, which becomes less accurate as the number of classes grows from 10 to 100. On the other hand, GPND does not use class label information and copes much better with the additional complexity induced by the increased number of classes.

Table 4: Comparison with ODIN [36]. ↑ indicates larger value is better, and ↓ indicates lower value is better.

| | Outlier dataset | FPR(95%TPR)↓ | Detection↓ | AUROC↑ | AUPR in↑ | AUPR out↑ |
|---|---|---|---|---|---|---|
| | | **ODIN-WRN-28-10 / ODIN-Dense-BC / GPND** | | | | |
| CIFAR-10 | TinyImageNet (crop) | 23.4/**4.3**/29.1 | 14.2/**4.7**/15.7 | 94.2/**99.1**/90.1 | 92.8/**99.1**/84.1 | 94.7/99.1/**99.5** |
| | TinyImageNet (resize) | 25.5/**7.5**/11.8 | 15.2/**6.3**/8.3 | 92.1/**98.5**/96.5 | 89.0/**98.6**/95.0 | 93.6/98.5/**99.8** |
| | LSUN (resize) | 17.6/**3.8**/4.9 | 11.3/**4.4**/4.9 | 95.4/**99.2**/98.7 | 93.8/**99.3**/98.4 | 96.1/99.2/**99.7** |
| | iSUN | 21.3/**6.3**/11.0 | 13.2/**5.7**/7.8 | 93.7/**98.8**/96.9 | 91.2/**98.9**/96.1 | 94.9/98.8/**99.7** |
| | Uniform | **0.0/0.0/0.0** | 2.5/2.5/**0.1** | 100.0/**99.9**/99.9 | **100.0/100.0/100.0** | 100.0/99.9/**99.5** |
| | Gaussian | **0.0/0.0/0.0** | 2.5/2.5/**0.0** | **100.0/100.0/100.0** | **100.0/100.0/100.0** | **100.0/100.0**/99.8 |
| CIFAR-100 | TinyImageNet (crop) | 43.9/**17.3**/33.2 | 24.4/**11.2**/17.2 | 90.8/**97.1**/89.1 | 91.4/**97.4**/83.8 | 90.0/96.8/**98.7** |
| | TinyImageNet (resize) | 55.9/44.3/**15.0** | 30.4/24.6/**9.5** | 84.0/90.7/**95.9** | 82.8/91.4/**94.6** | 84.4/90.1/**99.4** |
| | LSUN (resize) | 56.5/44.0/**6.8** | 30.8/24.5/**5.8** | 86.0/91.5/**98.3** | 86.2/92.4/**98.0** | 84.9/90.6/**99.6** |
| | iSUN | 57.3/49.5/**14.3** | 31.1/27.2/**9.3** | 85.6/90.1/**96.2** | 85.9/91.1/**95.6** | 84.8/88.9/**99.3** |
| | Uniform | 0.1/0.5/**0.0** | 2.5/2.8/**0.0** | 99.1/99.5/**100.0** | 99.4/99.6/**100.0** | 97.5/99.0/**99.7** |
| | Gaussian | 1.0/0.2/**0.0** | 3.0/2.6/**0.0** | 98.5/99.6/**100.0** | 99.1/99.7/**100.0** | 95.9/99.1/**100.0** |

Table 5: Comparison with baselines. All values are percentages. ↑ indicates larger value is better, and ↓ indicates lower value is better.

| | 10% | 20% | 30% | 40% | 50% | 10% | 20% | 30% | 40% | 50% | 10% | 20% | 30% | 40% | 50% |
|---|---|---|---|---|---|---|---|---|---|---|---|---|---|---|---|
| | **F1↑** | | | | | **AUROC↑** | | | | | **FPR(95%TPR)↓** | | | | |
| GPND | 98.2 | 97.1 | 96.1 | 95.0 | 93.9 | 98.1 | 98.0 | 98.0 | 98.0 | 98.0 | 8.1 | 9.1 | 8.7 | 8.8 | 8.9 |
| AE | 84.8 | 79.6 | 79.5 | 77.6 | 75.6 | 93.4 | 93.8 | 93.4 | 92.9 | 92.8 | 24.3 | 24.6 | 24.7 | 23.9 | 23.7 |
| P-VAE | 97.6 | 95.8 | 94.2 | 92.4 | 90.5 | 95.2 | 95.7 | 95.6 | 95.8 | 95.9 | 18.8 | 18.0 | 17.4 | 17.3 | 17.0 |
| P-AAE | 97.3 | 95.5 | 94.0 | 92.0 | 90.2 | 95.2 | 95.6 | 95.3 | 95.2 | 95.3 | 20.7 | 19.3 | 19.0 | 18.9 | 18.6 |
| | **Detection error↓** | | | | | **AUPR in↑** | | | | | **AUPR out↑** | | | | |
| GPND | 5.4 | 5.8 | 5.8 | 5.9 | 6.0 | 99.7 | 99.4 | 99.1 | 98.6 | 98.0 | 86.3 | 92.2 | 95.0 | 96.5 | 97.5 |
| AE | 11.4 | 11.4 | 11.6 | 12.0 | 12.2 | 98.9 | 97.8 | 95.8 | 93.2 | 90.0 | 78.0 | 86.0 | 89.7 | 92.0 | 94.0 |
| P-VAE | 9.8 | 9.7 | 9.7 | 9.7 | 9.5 | 99.3 | 98.7 | 97.8 | 96.7 | 95.6 | 81.7 | 89.2 | 92.5 | 94.6 | 96.3 |
| P-AAE | 9.4 | 9.3 | 9.5 | 9.8 | 9.8 | 99.2 | 98.6 | 97.4 | 96.0 | 94.3 | 79.3 | 87.7 | 91.5 | 93.7 | 95.4 |

## 6.3 Ablation

Table 5 compares GPND with some baselines to better appreciate the improvement provided by the architectural choices. The baselines are: i) vanilla AE with thresholding of the reconstruction error and same pipeline (AE); ii) proposed approach where the AAE is replaced by a VAE (P-VAE); iii) proposed approach where the AAE is without the additional adversarial component induced by the discriminator applied to the decoded image (P-AAE).

To motivate the importance of each component of $p_X(\bar{x})$ in (5), we repeat the experiment with MNIST under the following conditions: a) *GPND Complete* is the unmodified approach, where $p_X(\bar{x})$ is computed as in (5); b) *Parallel component only* drops $p_{W^\perp}$ and assumes $p_X(\bar{x}) = p_{W^\parallel}(\bar{w}^\parallel)$; c) *Perpendicular component only* drops $p_{W^\parallel}$ and assumes $p_X(\bar{x}) = p_{W^\perp}(\bar{w}^\perp)$; d) $p_Z(z)$ *only* drops also $|\det S^{-1}|$ and assumes $p_X(\bar{x}) = p_Z(z)$. dThe results are shown in Figure 5. It can be noticed that the scaling factor $|\det S^{-1}|$ plays an essential role in the *Parallel component only*, and that the *Parallel component only* and the *Perpendicular component only* play an essential role in composing the *GPND Complete* model.

Additional implementation details include the choice of hyperparameters. For MNIST and COIL-100 the latent space size was chosen to maximize $F_1$ on the validation set. It is 16, and we varied it from 16 to 64 without significant performance change. For CIFAR-10 and CIFAR-100, the latent space size was set to 256. The hyperparameters of all losses are one, except for $\mathcal{L}_{error}$ and $\mathcal{L}_{adv-d_z}$ when optimizing for $D_z$, which are equal to 2.0. For CIFAR-10 and CIFAR-100, the hyperparameter of $\mathcal{L}_{error}$ is 10.0. We use the Adam optimizer with learning rate of 0.002, batch size of 128, and 80 epochs.

## 7 Conclusion

We introduced GPND, an approach and a network architecture for novelty detection that is based on learning mappings $f$ and $g$ that define the parameterized manifold $\mathcal{M}$ which captures the underlying structure of the inlier distribution. Unlike prior deep learning based methods, GPND detects that a given sample is an outlier by evaluating its inlier probability distribution. We have shown how each architectural and model components are essential to the novelty detection. In addition, with a relatively simple architecture we have shown how GPND provides state-of-the-art performance using different measures, different datasets, and different protocols, demonstrating to compare favorably also with the out-of-distribution literature.

**Acknowledgments**

This material is based upon work supported by the National Science Foundation under Grant No. IIS-1761792.

## Footnotes

[1] `https://github.com/podgorskiy/GPND`

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
