[Reviews · NeurIPS 2018]

Reviewer 1



The problem of the study is outlier detection given only inlier training data. The high-level approach is to learn a density estimation function over the training data, and then filter out outliers using a learned threshold \gamma. They approximate the density function through a decomposition over the tangent space of the learned manifold near each given sample. To learn the manifold structure the authors use a variation of adversarial autoencoders. The evaluation is performed on MNIST, FashionMNIST, and COIL against a few baselines. Overall, the paper is very well-written and easy to follow -- the presentation progresses coherently. However, there are several issues that I think must be addressed before the work is published. At the heart of this work, the problem being addressed is that of density estimation. The learned function is then used to perform outlier detection. - There are several existing methods that are capable of performing density estimation. Autoregressive models such as PixelCNN, PixelCNN++, and the previous work, most of the recent compression techniques that use deep neural networks, and even a simple Variational Autoencoder can give us density estimates. There should be at least one of these density estimation methods to assess how well the proposed method approximates the density to determine whether the trouble of implementing such a complex technique is worth it. - The outlier detection problem is a quickly growing area which is not easy to track, but the authors should compare against easy-to-implement state-of-the-art methods such as ODIN [1] (ICLR 2018) at least. Thresholding the prediction of a predictive classifier, thresholding the reconstruction error of an AE that is trained with the same pipeline (without the additional loss functions) are two other easy to implement methods that they could compare against. - The presented evaluation is too weak. There are several measures in the outlier detection literature, from which the authors only report one or two. They don't adopt the same evaluation protocol either, yet, they use the numbers from other evaluations as a comparison. Furthermore, the evaluation is only performed on low-dimensional, low-complexity datasets MNIST, FashionMNIST, and COIL. There should be at least a CIFAR10 in the evaluation. It is common in the literature to also compare against CIFAR100 and downscaled imagenet. Quality. The submission is technically sound. The idea is interesting and the work is executed and presented well and coherently. However, as mentioned earlier, at a high-level, the direction of the work should be slightly corrected to better position it within the existing literature. Clarity. The paper is well-written and clear. The previous work could be significantly improved. In specific, I recommend adding the density estimation literature, and out-of-distribution detection literature as the method is addressing both of the problems. The mentioned work in the review could serve as a good starting point in these areas. The evaluation could also be improved as I discussed earlier. Originality. I haven't seen this idea executed before. It is both novel and interesting. It would be influential if it improves the density approximation problem or performs outlier detection better (shown more strongly in evaluation). Even if it does not outperform the other methods, there still needs to be a visible connection and adequate comparison to position the work properly in the literature. Significance. Unfortunately, I think, the work in its current state is unlikely to have an impact on the relevant areas. References. [1] Liang, Shiyu, Yixuan Li, and R. Srikant. "Enhancing the reliability of out-of-distribution image detection in neural networks." arXiv preprint arXiv:1706.02690 (2017). -=-=-=-=-=-=-=-=-=-=-=-=-=-=-=-=-=-=-=-=-=-=-=-=-= Update after rebuttal. I think the author response has adequately addressed the raised concerns -- the paper would be a good submission with the proposed revisions. I have increased the overall score to 7.

Reviewer 2



summary: This paper describes a probabilistic framework for anomaly detection using manifold learning by adversarial autoencoder. The data distribution is modeled as a smooth manifold with an independent noise component and the adversarial loss encourages the model to learn it by forcing the encoder to map x to a z with distribution close to N(0,1). A novelty test can then be derived from the product of the two independent probability density function components: one parallel to the manifold tangent space and one orthogonal to it (the noise). The paper shows that their approach provides state-of-the-art accuracy on 3 standard datasets (MNIST, COIL100, Fashion-MNIST). The novelty of this approach is to leverage recent developments in adversarial autoencoders into a probabilistic framework for anomaly detection. The independence assumptions made are reasonable and the principles used are sound. The results appear compelling, at least for the datasets used in the experiments. Those datasets are relatively clean with nice separation between classes. It remains to be seen how well this approach extends to noisier real-life data. The quality of the paper is ok. There are many grammatical errors (see below). The organization of the text is generally good, however i wished more emphasis would have been put into an intuitive interpretation of the approach in a discussion/conclusion section. In particular the conclusion is inadequate. Also there is no discussion about the computational complexity of the approach. Some details are lacking, such as the architecture used for the autoencoders. In general, it seems the quality of the paper is degrading substantially towards the end. The math is well laid out and relatively easy to follow. I found only one notation error (see below). It think the algorithm should be reasonably easy to implement and the results should be reproducible. Overall, i think this is a worthwhile paper for NIPS. It present a novel approach that can potentially be of interest to many practitioners. The technical quality is high despite some small issues. Details: - Table 1 is missing the definition of the value (is it F1, AUC ?). - Table 2 is missing the reference [25] for L1-thresholding - The architecture of the autoencoder is not given. At least the size of the bottleneck should be given. - Equation (4): pW(w,w_|) should be pW(w||,w_|) - Line 7: likely is -> likely it is - Line 12: we improved -> we improve - Line 39: generating -> generate - Line 82: leanning -> learning - Line 85: it represents -> we represent - Line 85: sore -> score - Line 101: we have that -> we have - Line 245: Because count -> Because the count - Line 247: on training set -> on the training set - Line 249: since in [8] is used pretrained VGG -> since in [8], a pretrained VGG is used - Line 252: before uses pretrained -> before, uses a pretrained - Line 252: we train autoencoder -> we train an autoencoder - Line 253: on very limited number samples -> on a very limited number of samples - Line 257: we repeat experiment -> we repeat the experiment - Line 261: We drop perpendicular -> we drop the perpendicular - Line 266: because curve -> because the curve - Line 266: only performs significantly better -> performs significantly better - line 267: better than "pz(z) only" one -> better than the "pz(z)"-only one - Line 267: plays essential role -> plays an essential role - Line 269: presented a approach -> presented an approach - Line 274: Our analysis showed that the method shows -> Our method shows ======================= Update after author's feedback ======================== I appreciate that the authors promise to expand comparisons and add details on computational complexity. My overall score remains unchanged.

Reviewer 3



Summary: The paper proposes a probabilistic approach to detect whether a new data point is an outlier or an inlier with respect to the training data manifold. The proposed approach, called Generative Probabilistic Novelty Detection (GPND), utilizes an adversarially trained autoencoder. The decoder network defines a manifold in the observation space and the encoder network projects the data onto the manifold. An adversarial loss is defined to shape the latent space to be a known parametric distribution. A reconstruction error loss and an adversarial loss are defined to ensure the reconstruction quality of the projected inlier data. The training data (inlier) is modeled as being sampled from the manifold with additive noise and the distribution is modeled via the product of (a) the distribution in the tangent space which is a function of the target distribution in the latent space, and, (b) The distribution of the residuals in the normal space which is considered stationary. A sample is considered an output if it’s probability w.r.t the inlier distribution is below a threshold. The paper validates the approach by setting up (artificial) outlier detection tasks using the MINST, COIL-100 and Fashion-MNIST datasets. The experiments on these datasets show that this probabilistic estimation of outliers works better than the existing approaches. Strengths: The paper addresses an important problem – how to detect an outlier or an example atypical of the training data. This problem is significant and important for deploying any ML system in an open-world scenario. The authors present an incrementally novel approach – using adversarially trained (both latent space and output) autoencoders with a probabilistic formulation to calculate the probability that any sample is from the inlying distribution. I haven’t seen the specific derivations before although they seem straightforward. The exposition is mostly clear and the approach is technically sound. Results on MNIST, COIL-100 and Fashion-MNIST datasets seem satisfactory. Ablation studies are also performed to quantify the impact of the factors in p(w). Weaknesses: I have a few points below which I hope can be addressed in the rebuttal. 1. Choice of certain hyperparameters – how is the size of the latent space chosen? 2. Experimental results: Ablation study: needed to understand the impact of different components of the loss function – reconstruction loss and the two adversarial losses – one for the latent space, other for the output as also the impact of the probabilistic model. The authors allude to the fact that adversarial training improves the model, this is not experimentally validated anywhere. Manifold projection and the residual: The authors model the two probability distributions – one on the manifold via the latent space distribution and the other for the residual of the projection onto the manifold. Independence and stationarity are assumed to compute the two and use the product as the probability for generating an inlier data point. Since outliers are expected to be away from the manifold, the projection residual is expected to play a bigger role in identifying outliers. This is not the case as shown in Figure 5. Why? Please discuss. Comparison with other methods: There is also no comparative discussion of their results vis-à-vis the state of the art.